# Neutralizing Antibodies against SARS-CoV-2: Importance of Comorbidities in Health Personnel against Reinfections

**DOI:** 10.3390/v15122354

**Published:** 2023-11-30

**Authors:** Cruz Vargas-De-León, Mónica Alethia Cureño-Díaz, Ma. Isabel Salazar, Clemente Cruz-Cruz, Miguel Ángel Loyola-Cruz, Emilio Mariano Durán-Manuel, Edwin Rodrigo Zamora-Pacheco, Juan Carlos Bravata-Alcántara, Gustavo Esteban Lugo-Zamudio, Verónica Fernández-Sánchez, Juan Manuel Bello-López, Gabriela Ibáñez-Cervantes

**Affiliations:** 1División de Investigación, Hospital Juárez de México, Ciudad de México 07760, Mexico; leoncruz82@yahoo.com.mx (C.V.-D.-L.); cruzc1990@hotmail.com (C.C.-C.); miguelqbp@gmail.com (M.Á.L.-C.); emilioduranmanuel@gmail.com (E.M.D.-M.);; 2Laboratorio de Modelación Bioestadística para la Salud, Sección de Estudios de Posgrado e Investigación, Escuela Superior de Medicina, Instituto Politécnico Nacional, Ciudad de México 07738, Mexico; rodrigo.zampac@gmail.com; 3Dirección de Investigación y Enseñanza, Hospital Juárez de México, Ciudad de México 07760, Mexico; dracureno@yahoo.com.mx; 4Laboratorio Nacional de Vacunología y Virus Tropicales, Escuela Nacional de Ciencias Biológicas (ENCB), Instituto Politécnico Nacional, Ciudad de México 11340, México; isalazarsan@yahoo.com; 5Laboratorio de Diagnóstico Molecular, Hospital Juárez de México, Ciudad de México 07760, Mexico; vaio_df@hotmail.com; 6Dirección General, Hospital Juárez de México, Ciudad de México 07760, Mexico; zalugusta1@gmail.com

**Keywords:** neutralizing antibodies, SARS-CoV-2, COVID-19, vaccination, healthcare workers, reinfections

## Abstract

One of the priority lines of action to contain the SARS-CoV-2 pandemic was vaccination programs for healthcare workers. However, with the emergence of highly contagious strains, such as the Omicron variant, it was necessary to know the serological status of health personnel to make decisions for the application of reinforcements. The aim of this work was to determine the seroprevalence against SARS-CoV-2 in healthcare workers in a Mexican hospital after six months of the administration of the Pfizer-BioNTech vaccine (two doses, 4 weeks apart) and to investigate the association between comorbidities, response to the vaccine, and reinfections. Neutralizing antibodies against SARS-CoV-2 were determined using ELISA assays for 262 employees of Hospital Juárez de México with and without a history of COVID-19. A beta regression analysis was performed to study the associated comorbidities and their relationship with the levels of antibodies against SARS-CoV-2. Finally, an epidemiological follow-up was carried out to detect reinfections in this population. A significant difference in SARS-CoV-2 seroprevalence was observed in workers with a history of COVID-19 prior to vaccination compared to those without a history of the disease (MD: 0.961 and SD: 0.049; <0.001). Beta regression showed that workers with a history of COVID-19 have greater protection compared to those without a history of the infection. Neutralizing antibodies were found to be decreased in alcoholic and diabetic subjects (80.1%). Notably, eight cases of Omicron reinfections were identified, and gender and obesity were associated with the presence of reinfections (6.41 OR; 95% BCa CI: 1.15, 105.0). The response to the vaccine was influenced by the history of SARS-CoV-2 infection and associated comorbidities. The above highlights the importance of prioritizing this segment of the population for reinforcements in periods of less than one year to guarantee their effectiveness against new variants.

## 1. Introduction

The pandemic caused by the SARS-CoV-2 coronavirus began in December 2019 in Wuhan, Hubei province of China, and subsequently spread throughout the world [1,2]. The emergence of new highly infectious variants has led to waves and cases of reinfection, making it difficult to reduce the number of cases [2]. On 11 March 2020, the World Health Organization (WHO) classified the outbreak as a pandemic, resulting in 770,875,433 confirmed cases and 6,959,316 deaths as of 19 September 2023 [3]. The high incidence and mortality of COVID-19 was the reason for the start of intensive work on the development of an effective vaccine [4,5,6]. In Mexico, the vaccination process against this disease began with the population over 65 years of age and with healthcare workers at the end of December 2020 (in 2023, there are 81,849,962 Mexicans vaccinated). Frontline healthcare workers face a substantial risk of SARS-CoV-2 infection due to close contact with confirmed patients or exposure to undiagnosed or subclinical infectious cases [7]. Research reports many healthcare workers infected with SARS-CoV-2 worldwide [8,9,10,11,12]. This increase in the number of cases among healthcare workers includes not only doctors and nurses directly caring for patients with COVID-19 but also orderlies, chemists, and administrative staff. When healthcare workers become ill with COVID-19, they are unable to work or provide key services to patients, so having staff protected through vaccination is a priority action. Given the evidence of the high risk of SARS-CoV-2 infection among healthcare workers and their critical role during the pandemic [13,14], protecting them against this disease has been a national and international priority. Thus, early access to the COVID-19 vaccine for healthcare workers was crucial to ensuring the safety of this essential workforce. Knowledge of the human antibody response generated by the SARS-CoV-2 vaccination process can contribute to new vaccine development and strategies to guide the design, implementation, and interpretation of serological assays for surveillance purposes [15,16]. The aim of this work was to determine the importance of pre-existing comorbidities and their influence on the production of neutralizing antibodies against SARS-CoV-2 in healthcare workers from a Mexican hospital six months after the administration of the Pfizer-BioNTech vaccine, the vaccine response, and the prevalence of reinfection. The need for booster vaccination in healthcare workers is analyzed and discussed, emphasizing the priority of people with comorbidities.

## 2. Material and Methods

### 2.1. Study Population

The participants in this present study were healthcare workers from different services of the Hospital Juárez de México, which was destined for the care of COVID-19 patients. Participants were subjects vaccinated with the full schedule (two doses, 4 weeks apart) in January 2021, with the Pfizer-BioNTech vaccine. So, 6 months after full vaccination, baseline demographic data, comorbidities, and the history of COVID-19 before and after the vaccination process were collected. Only workers with a full dose of vaccination within the first 6 months after vaccination were included. Workers vaccinated with other brands and those who were under home protection during the study analysis were excluded. Two groups were formed: (A) with no history of SARS-CoV-2 infection and (B) with a history of SARS-CoV-2 infection, both prior to vaccination. In group B subjects, infection was confirmed through real-time reverse transcription-polymerase chain reaction (RT-PCR) according to the Berlin protocol [17].

### 2.2. Detection of Neutralizing Antibodies against SARS-CoV-2

To obtain serum from each participant, 15 mL of whole blood was drawn into a tube with EDTA anticoagulant. The sample was transported to the laboratory for analysis and processed in a time not exceeding 45 min. Then, the samples were centrifuged at 2100× *g* for 5 min. The fraction was obtained after centrifugation corresponding to the blood serum, which was stored at −70 °C until use. The detection of IgG neutralizing antibodies against SARS-CoV-2 (antibodies fraction) was determined through ELISA assays using the “SARS-CoV-2 Neutralization Antibody Detection Kit” (GenScript, REF: L00847), following the manufacturer’s instructions. The SARS-CoV-2 Neutralization Antibody Detection Kit detects circulating neutralizing antibodies (IgG neutralizing antibodies) against SARS-CoV-2 that block the interaction between the receptor binding domain (RBD) of the spike glycoprotein viral and the cell surface receptor ACE2 (the neutralizing test is not capable of detecting antibodies against circulating CoVs during the time of the study and its follow-up). For the interpretation of the tests, results were considered positive when values equal to or above 30% neutralizing antibodies were identified (cut-off point below 30%) [18]. The inhibition rate was calculated as follows: percentage of signal inhibition = [1 − (OD value of sample/OD value of negative control) × 100].

### 2.3. Epidemiological Monitoring for the Identification of Reinfection

The study population was under epidemiological surveillance to identify reinfection events. According to the WHO, the operational definition of reinfection is a suspected or probable case of SARS-CoV-2 reinfection, plus laboratory studies with comparative genomic analysis of SARS-CoV-2 virus from the primary and secondary samples showing evidence that they belong to different genetic clades or lineages [19]. For this purpose, participants with symptoms suggestive of COVID-19 were subjected to rapid antigen detection tests. Participants who tested positive for the rapid test were subjected to the identification of SARS-CoV-2 and its variants by using the qPCR kit “MASTER MUT Omicron configuration” (Genes2Life, Irapuato Guanajuato, Mexico), which is a qPCR kit that allows for the detection of SARS-CoV-2 variants, such as Alpha (α), Beta (β), Epsilon (ε), Eta (η), Kappa (κ), and Lambda (λ) using a single marker, and Delta (δ) and Omicron (ο) BA.1 and BA.2 using more than one marker. In RT-PCR assays, positive controls for the *E*, *RdRp*, and *RNAse P* genes were provided by the “Instituto de Diagnóstico y Referencia Epidemiológicos” (InDRE-Mexico).

### 2.4. Statistical Analysis

Data are presented using mean (standard deviation, SD) and counts (percentage) for quantitative and qualitative variables, respectively. Based on the history of SARS-CoV-2 infection prior to vaccination, two groups were formed: those with a history of infection and those with no history of infection. Quantitative variables were compared by using Student’s *t*-test, and qualitative variables were compared by using the Chi-square test with or without Yates’s corrections.

The beta distribution of the proportion SARS-CoV-2 neutralizing antibodies fraction was checked by using the descdist function of the R fitdistrplus library [20]. Beta regressions were performed with the betareg function of the R betareg library [21]. We define the saturated model as the one with all the design variables and the minimal model as the parsimonious model that fits as well. The saturated model contained the following regressors: age, sex, comorbidities (diabetes, obesity, and hypertension), addictions (alcoholism and smoking), and SARS-CoV-2 infection. The response variable is the proportion of SARS-CoV-2 neutralizing antibodies fraction. The cauchit link function was used in the models implemented. Models that include a variable dispersion parameter (phi), i.e., they consider that the dispersion parameter depends on a set of regressors through a log link function by using the Akaike’s information criterion (AIC) and the likelihood-ratio test to select the best model. The criteria for the fit of the selected model were checked. The beta regressions of the minimal model were used to predict the percentage of antibodies against SARS-CoV-2 according to the selected variables.

Groups with and without reinfection were compared by using the Chi-square test with or without Yates’s corrections for qualitative variables and Student’s *t*-test with 1000 bootstrap samples. Logistic regression analysis was used for multivariate analysis to explore associations of reinfections with demographic variables, comorbidities, and addictions. Given the low proportion of reinfections, logistic regression parameters were estimated with the penalized likelihood method by using the logistf function from the R logistf library [22]. An interval estimate for odds ratios (ORs) was obtained from 1000 simple bootstraps by using the boot function from the R “boot” library [23,24]. We estimated 95% bias-corrected and accelerated (BCa) bootstrap confidence intervals for the OR for the demographic variables, comorbidities, and addictions.

*p* values < 0.05 were considered statistically significant. Analyses were performed with the R software, version 3.4.4, and the box plot was produced in GraphPad Prism 8.4.0.

### 2.5. Ethical Aspects

The institutional Committee of Research, Ethics, and Biosafety from Hospital Juárez de México (HJM) approved the protocol under the registration number HJM16/21-I “Detection of anti-SARS-CoV-2 neutralizing antibodies in the working population of the Hospital Juárez de Mexico” in accordance with the Regulation of the General Health Law on Research for Health [25].

## 3. Results

### 3.1. Description of the Study Population

A total of 262 healthcare workers from the *Hospital Juárez de México* were included in this study. One hundred fifteen participants reported previous SARS-CoV-2 infection (confirmed through RT-PCR) prior to the vaccination process. Most of the population was female (*n* = 193/73.7%). The mean age was 44.2 years (SD 10.5). The most frequent comorbidities were hypertension (14.9%), type 2 diabetes mellitus (DM) (7.25%), and obesity (4.19%). The most frequent addictions were smoking (10.7%) and alcoholism (8.79%). Demographic variables, comorbidities, and addictions in both groups were homogeneous, particularly for age, sex, hypertension, DM, obesity, smoking, alcoholism, asthma, and thyroid diseases. The most frequent comorbidities among workers with a history of infection prior to vaccination were hypertension (13.9%), DM (7.8%), obesity (5.2%), smoking (7.0%), and alcoholism (7.8%) (Table 1).

### 3.2. Detection of Neutralizing Anti-SARS-CoV-2 Antibodies

Regarding the percentage of anti-SARS-CoV-2 antibodies in the study population, a significant difference (*p* < 0.001) was observed in workers who had a history of SARS-CoV-2 infection prior to vaccination compared to workers with no history of infection (Figure 1 and Table 2).

### 3.3. Variable Dispersion Beta Regression

Variable dispersion beta regression models were calculated to predict the proportion of SARS-CoV-2 IgG antibodies, as shown in Table 3.

The saturated model is based on comorbidities, addictions, and the history of SARS-CoV-2 infection, and is adjusted for age and sex. The saturated model has an AIC of −974.6. We selected the more parsimonious minimal beta regression model with an AIC of −981.3, which only considers the regressor variables of diabetes, alcoholism, and history of SARS-CoV-2 infection. The saturated and minimal models assume that the dispersion parameter is not constant for all workers under consideration, which is a more realistic assumption. On the one hand, the statistically significant regressors of the model were DM (*p* = 0.009), alcoholism (*p* = 0.040), and SARS-CoV-2 infection (*p* < 0.001). A positive coefficient of the beta regression, 4.37 (95% CI: 3.16, 5.58), means that workers with a history of SARS-CoV-2 infection have a higher proportion of SARS-CoV-2 IgG antibodies than workers without a history of infection. Conversely, a negative coefficient of −1.73 (95% CI: −3.04, −0.417) means that workers with DM have a lower proportion of SARS-CoV-2 IgG antibodies than workers without DM. Similarly, the negative coefficient for alcoholism, −1.06 (95% CI: −2.08, −0.048), means that workers with alcoholism have a lower proportion of SARS-CoV-2 IgG antibodies than workers without alcoholism. Table 4, which shows the predictions of SARS-CoV-2 IgG antibody percentages by using the minimal model, shows that patients who had SARS-CoV-2 infection prior to vaccination have higher antibody percentages than patients who were not infected with SARS-CoV-2.

The model predicts that patients with diabetes, alcoholism, and no SARS-CoV-2 infection prior to vaccination have 80.1% SARS-CoV-2 IgG antibodies, in contrast to how many patients without diabetes, without alcoholism, and who had SARS-CoV-2 infection prior to vaccination have neutralizing antibodies (96.3%). Alternatively, as the name suggests, the variable dispersion beta regression models that were fitted allow for the value of the dispersion parameter to vary between individuals. The dispersion parameter has an inverse relationship with the variance of the response variable. Consequently, in both the saturated model and the minimal model, we observed that the variation in SARS-CoV-2 IgG antibody levels between individuals is explained by whether they have SARS-CoV-2 infection and whether they have the comorbidities DM and obesity.

### 3.4. Reinfection after Vaccination

In the follow-up, 18 reinfections of 262 with SARS-CoV-2 were detected after vaccination, and 8 of these were reinfections with Omicron. These 8 reinfections out of a total of 262 occurred in women, 6 of these reinfections were over 40 years of age, 1 was a smoker, and one was obese. We found no differences in age, comorbidities, and addictions between the reinfected and non-reinfected groups (Table 5).

We found that men have lower risks (0.172 OR; 95% BCa CI: 0.047, 0.433) than women for reinfection with SARS-CoV-2, and that subjects with obesity (6.41 OR; 95% BCa CI: 1.15, 105.0) have higher risks of reinfection than non-obese subjects. No association was found between the percentage of anti-SARS-CoV-2 antibodies, other comorbidities, and addictions with the presence of reinfection (Table 6).

## 4. Discussion

Due to the presence of cases of reinfection, the emergence of SARS-CoV-2 variants, and the difference in vaccination rates worldwide, it is essential to know the duration of protection after natural infection or vaccination-acquired immunity. In this study, we analyzed the presence of antibodies six months post-vaccination in healthcare workers at the Hospital Juárez de México who had or had not previously been infected with SARS-CoV-2 and whether they had reinfection post-vaccination. It was observed that the percentage of anti-SARS-CoV-2 antibodies in workers who had a history of SARS-CoV-2 infection prior to vaccination was higher compared to workers with no history of infection. The development of an immune response against SARS-CoV-2 induced by the infectious process has crucial implications for reinfection and vaccine effectiveness [26]. The presence of comorbidities influences antibody generation; we found that patients with diabetes and alcoholism and who were not infected with SARS-CoV-2 before vaccination had 80.1% IgG antibodies against SARS-CoV-2. In contrast, patients without diabetes, without alcoholism, and who were infected with SARS-CoV-2 before vaccination had higher protection (96.3%). Previous studies suggest that infection provides natural immunity for at least 3 months [27], and immunity remains stable for up to 6–8 months after the initial infection [28,29,30]. However, the WHO indicates that the presence of antibodies in recovered patients does not guarantee protection against reinfection [31]. We speculate that previous SARS-CoV-2 infection induces detectable immune responses in most reported cases; therefore, the vaccination process generates greater protection against possible reinfection events. We tested this hypothesis by detecting higher percentages of protection in participants with a previous history of infection. Post-infection immunity is known to be generated by the immune cell-mediated humoral response [32]. Regarding reinfection, the first observed cases of SARS-CoV-2 reinfection were reported in Hong Kong in unvaccinated patients with mild symptoms for the first and no symptoms for the second infection, with 142 days between two episodes [33]. In this study, we found that men are at lower risk than women for reinfection with SARS-CoV-2; this phenomenon has already been reported in other studies [34]. We observed that women have a significantly lower percentage of SARS-CoV-2 antibodies than men. Differences in susceptibility and response between men and women to viral infections are known to exist, resulting in differences in disease incidence and severity [32]. Studies have shown that women are less susceptible to viral infections due to reduced cytokine production as well as increased activity of macrophages, neutrophils, and increased antibody production [35,36,37]. Women are less likely to produce extreme immune responses to bacterial or viral infections than men [36]. Women’s protection against microbial and viral conditions is provided by the X chromosome and sex hormones that modulate innate and adaptive immunity [38]. Male patients have been observed to have higher circulating levels of TNF-α than female patients, which correlates with a worse prognosis [39,40]. This exacerbated response in men may be related to a higher production of neutralizing antibodies and a longer-lasting immune response compared to women, which confers protection against reinfection. According to the clinical characteristics of the participants who were vaccinated and experienced reinfection events, they had mild symptoms. This phenomenon has already been observed in a previous work, where they reported that reinfection events occurred in vaccinated individuals despite the presence of antibodies against SARS-CoV-2 [41]. Statistics analysis has shown that the reinfection rate in vaccinated individuals is between less than 0.5% and more than 5% [36,42]. Regarding the presence of comorbidities and reinfection with SARS-CoV-2, we found that obese subjects have a higher risk of reinfection than non-obese subjects. Obesity is an inflammatory state associated with chronic activation of the immune system, affecting proper immune functions and host defense mechanisms, leading to high complication rates in infectious diseases and higher rates of vaccine failure [42,43]. Obesity has been considered a risk factor for various infections, with post-infection complications and increased mortality from serious infections [44]. Obesity has been shown to have deleterious effects on host immunity, mainly increasing the risk of infectious susceptibility and severity [45,46]. Furthermore, for the other comorbidities and addictions analyzed, no association was found between the percentage of anti-SARS-CoV-2 antibodies and the presence of reinfection. The results obtained here show that there is substantial variation between individuals in the immune response to vaccination, both in quantity and quality. We note that high-risk populations, including the elderly, people with obesity, and people with comorbidities such as type 2 diabetes mellitus, are more susceptible to increased disease severity and decreased vaccine efficacy. Likewise, these high-risk populations present modifications in their microenvironments and unique immune responses that contribute to greater vulnerability to infections. This study shows the importance of developing policies focused on immune surveillance after a vaccination process in order to evaluate the effectiveness of the vaccine, monitor the immune response, and contribute to the protection or control of this and other infections. Further studies over longer periods of time are needed to monitor the behavior of antibody production against the new variants and to consider the administration of vaccine mixtures for booster doses. The significance of our study is to emphasize the need to prioritize vaccination for people with diabetes and obesity, followed by other comorbidities, as well as to raise awareness of the effect of smoking on the immune system. One of the limitations of this study is associated with participants who did not report previous SARS-CoV-2 infection and were not tested using the standard PCR test, and therefore, may have been asymptomatic.

## 5. Conclusions

The persistence of SARS-CoV-2 neutralizing IgG antibodies depends on previous infection with SARS-CoV-2, as well as on the presence of comorbidities such as obesity and diabetes mellitus. The results obtained here show that obese subjects have a higher risk of reinfection, so obesity directly influences the inflammatory state associated with the chronic activation of the immune system, which affects adequate immune functions and host defense mechanisms, leading to high rates of infectious disease complications and higher vaccine failure rate. Furthermore, long-term studies are needed to determine the duration of protection against reinfection with novel variants of the virus in survivors and to determine whether individuals with asymptomatic infection are at increased risk of reinfection.

## Figures and Tables

**Figure 1 viruses-15-02354-f001:**
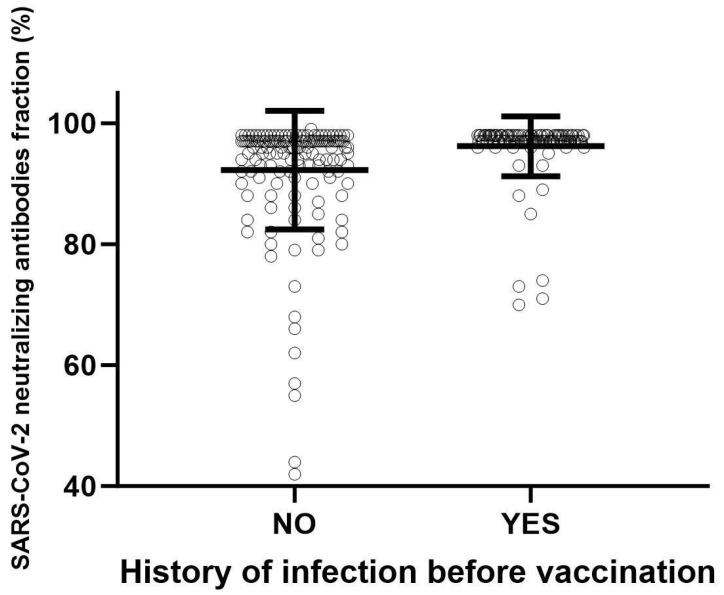
Percentage (%) of SARS-CoV-2 neutralizing antibodies fraction in the study population. A significant difference (*p* < 0.001) was observed in workers who had a history of SARS-CoV-2 infection prior to vaccination compared to workers with no history of infection. Student’s *t*-test was used for the comparison of two means.

**Table 1 viruses-15-02354-t001:** Demographic and comorbidities characteristics of a total of 262 workers vaccinated at the Hospital Juárez de México.

Clinical Characteristics	History of Infection before Vaccination		*p*-Value
	Present n = 115	Absent n = 147	
Age (Years) *	43.8 (10.6)	44.5 (10.4)	0.570 ^§^
Age (≥40 years) **	75 (65.2%)	103 (70.1%)	0.404 ‡
Sex (Male) **	31 (27.0%)	38 (25.9%)	0.840 ‡
Diabetes **	9 (7.8%)	10 (6.8%)	0.751 ‡
Obesity **	6 (5.2%)	5 (2.4%)	0.677 †
Hypertension **	16 (13.9%)	23 (15.6%)	0.696 ‡
Alcoholism **	9 (7.8%)	19 (12.9%)	0.185 ‡
Smoking **	8 (7.0%)	15 (10.2%)	0.357 ‡
Asthma **	6 (5.2%)	4 (2.7%)	0.477 †
Thyroid diseases **	5 (4.3%)	5 (3.4%)	0.943 †

* Mean (SD); SD, Standard Deviation. ** Absolute frequency (n) and percentage (%). ^§^ Student’s *t*-test. ‡ Chi-square test. † Chi-square test with Yates’s corrections.

**Table 2 viruses-15-02354-t002:** Proportion of SARS-CoV-2 neutralizing antibodies fraction in vaccinated workers at the Hospital Juárez de México.

	History of Infection before Vaccination				
**Proportion of SARS-CoV-2 neutralizing antibodies fraction**	**Present** **n = 115**		**Absent** **n = 147**		***p*-Value**
	Mean (SD)	Min, Median, Max	Mean (SD)	Min, Median, Max	
	0.961 (0.049)	0.700, 0.970, 0.980	0.922 (0.097)	0.420, 0.960, 0.990	**<0.001 ***

* Student’s *t*-test.

**Table 3 viruses-15-02354-t003:** Effect of comorbidities, addictions, and SARS-CoV-2 infection history on the proportion of SARS-CoV-2 antibodies in vaccinated workers at the Hospital Juárez de México.

Clinical Characteristics	Saturated Model *				Minimal Model **			
	B	95% IC		*p*-Value	B	95% IC		*p*-Value
		LL	LU			LL	LU	
**Intercept**	**5.135**	**3.099**	**7.170**	**<0.001**	4.18	3.53	4.83	**<0.001**
Age (Years)	−0.026	−0.068	0.016	0.224				
Sex (Male)	0.562	−0.468	1.59	0.285				
Diabetes	−1.65	−2.97	−0.319	**0.015**	−1.73	−3.04	−0.417	**0.009**
Obesity	1.06	−1.12	3.24	0.342				
Hypertension	0.395	−0.878	1.67	0.543				
Alcoholism	−1.29	−2.46	−0.124	**0.030**	−1.06	−2.08	−0.048	**0.040**
Smoking	−0.022	−1.26	1.21	0.971				
COVID-19 infection history	4.28	3.07	5.50	**<0.001**	4.37	3.16	5.58	**<0.001**
Phi coefficients (Dispersion)								
Phi (Intercept)	2.66	2.41	2.91	**<0.001**	2.65	2.41	2.90	**<0.001**
Phi (Diabetes)	−0.800	−1.46	−0.137	**0.018**	−0.893	−1.56	−0.228	**0.008**
Phi (Obesity)	1.04	0.164	1.93	**0.020**	0.846	0.029	1.66	**0.042**
Phi (COVID-19 infection history)	1.27	0.906	1.63	**<0.001**	1.29	0.932	1.65	**<0.001**

**B**: regression coefficient; **SE**: standard error; **95% CI**: 95% confidence interval; **LL**: lower limit; **LU**: upper limit. Significant values are in bold. * The saturated model contained the following regressors: age, sex, diabetes, obesity, hypertension, alcoholism, smoking, and SARS-CoV-2 infection. ** The minimal model contained the following regressors: age, diabetes, alcoholism, and SARS-CoV-2 infection. Beta regressions were performed to test the association between factors and the proportion of SARS-CoV-2 antibodies.

**Table 4 viruses-15-02354-t004:** A complex contingency table involving three dichotomous variables with the prediction of the percentage of SARS-CoV-2 antibodies using the minimal model.

				History of Infection before Vaccination	
				Present	Absent
**Diabetes**	Present	**Alcoholism**	Present	94.5%	80.1%
			Absent	95.4%	87.7%
	Absent	**Alcoholism**	Present	95.8%	90.1%
			Absent	**96.3%**	92.5%

Beta regressions of the minimal model were used to predict the percentage of SARS-CoV-2 antibodies according to the following conditions: diabetes, alcoholism, and history of infection before vaccination.

**Table 5 viruses-15-02354-t005:** Demographic and comorbidities characteristics of the reinfection with Omicron after vaccination.

Clinical Characteristics	Reinfection with Omicron after Vaccination		Bootstrap *p*-Value
	Present n = 8	Absent n = 254	
**Age (Years) ***	43.1 (9.28)	44.2 (10.5)	0.747 ^§^
**Age (40 years and more)**	6 (75.0%)	172 (67.7%)	0.667 †
**Sex (Male) ****	0 (0.0%)	69 (27.2%)	0.232 †
**Diabetes ****	0 (0.0%)	19 (7.5%)	0.885 †
**Obesity ****	1 (12.5%)	10 (3.9%)	0.879 †
**Hypertension ****	0 (0.0%)	39 (15.4%)	0.558 †
**Alcoholism ****	0 (0.0%)	28 (11.0%)	0.832 †
**Smoking ****	1 (12.5%)	22 (8.7%)	0.962 †

* Mean (SD); SD, Standard Deviation. ** Absolute frequency (n) and percentage (%). **^§^** Student’s *t*-test. † Chi-square test with Yates’s corrections.

**Table 6 viruses-15-02354-t006:** Effect of comorbidities, addictions, and percentage of SARS-CoV-2 antibodies on the reinfection by SARS-CoV-2.

Clinical Characteristics	OR	95% BCa Bootstrap IC	
		LL	LU
Age (Years)	1.000	0.907	1.09
Sex (Male)	**0.172**	**0.047**	**0.433**
Diabetes	0.952	0.010	3.67
Obesity	**6.41**	**1.15**	**105.0**
Hypertension	0.399	0.122	1.34
Alcoholism	0.634	0.167	2.024
Smoking	3.25	0.571	33.8
Percentage of SARS-CoV-2 antibodies	1.03	0.970	1.75

**OR**: odds ratio; **95% CI**: 95% confidence interval; **BCa**: bias-corrected and accelerated; **LL**: lower limit; **LU**: upper limit. Significant values are in bold. Logistics regressions were performed to test the association between factors and the reinfection by SARS-CoV-2.

## Data Availability

All data generated or analyzed during this study are included in this published article.

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
