# Peer review of "Neutralizing Antibodies against SARS-CoV-2: Importance of Comorbidities in Health Personnel against Reinfections"

_viruses, 2023, doi:10.3390/v15122354_

Round 1

Reviewer 1 Report (Previous Reviewer 3)

Comments and Suggestions for Authors

-        The attached English Certificate of editing does not correspond in title to the submitted manuscript.

-        Define the number of vaccination of participants to the study in the abstract.

-        The abstract need to be implemented with more results obtained in the study, and avoid to describe them generally.

-        The assay for neutralization test is a surrogate virus neutralization assay, more informative that ELISA, reported in the text.

-        References needs to be updated to 2023. In the whole manuscript the most recent publication was published in 2022.

-        I suppose that data of antibodies represents the inhibition binding ACE2/RBD ant not the “IgG percentage”. This important point must be revised in all sections of the work.

-        The first sentence of the introduction is uncomplete (lines 43-44).

-        Lines 47-47: it’s important to specify the year of reference cases.

-        Line 52: “currently” is ambiguous. It’s important to specify everything!

-        The full schedule of vaccination is not based on only two doses, at the moment! Revise the points in the whole manuscript.

-        Define the interval between the two vaccine doses received.

-        Convert data expressed in “rpm” to “g”.

-        Include reference for the interpretation of the limit of positive neutralization.

-        Define “antibody fraction” in the text and in y axis of fig. 1.

-        The use of asterisks in Tab 1 and Tab 5 is not clear for the reader.

-        The statistical methods used in Tab and Fig needs to be specified in the description of Tab/Fig.

-        Define clearly in the text the meaning of saturated and minimal model.

-        Tab 4 is difficult to understand for the three variables reported. It’s important to clarify how the percentage of prediction were determined.

-        The discussion need to be stressed on the presence of comorbidities in the vaccination response against SARS-CoV-2 infection of other possible infections, since this is the topic reported in the title of the work.

Comments on the Quality of English Language

Minor editing of English language required.

Author Response

We sent the responses to the comments of the reviewers of the manuscript entitled: “Neutralizing antibodies against SARS-CoV-2: importance of comorbidities in health personnel against reinfections with ID 2666540” each reviewer's changes were marked in the manuscript in yellow.

Reviewer 2 Report (Previous Reviewer 2)

Comments and Suggestions for Authors

The manuscript is a resubmission, and results improved as the authors addressed almost all the reviewers' comments. However, there is still some information that is missing.

- The authors must explain what is the target of the neutralizing antibodies detected, if the S or N protein, or both. Moreover, if the neutralizing test is able to detect antibodies against the VOCs circulating during the time of the study and its follow-up.

- The multivariate analysis should be added, if possible, to have a general view of the collected data and vaccinees' status.

Author Response

We sent the responses to the comments of the reviewers of the manuscript entitled: “Neutralizing antibodies against SARS-CoV-2: importance of comorbidities in health personnel against reinfections with ID 2666540” each reviewer's changes were marked in the manuscript in yellow.

Reviewer 3 Report (Previous Reviewer 1)

Comments and Suggestions for Authors

please correct any misspellings in the context. 

see attachment below.

The revised version answered my previous questions, and therefore I would be happy to accept the paper after minor revision on formatting.

Comments on the Quality of English Language

see attachment.

Author Response

We sent the responses to the comments of the reviewers of the manuscript entitled: “Neutralizing antibodies against SARS-CoV-2: importance of comorbidities in health personnel against reinfections with ID 2666540” each reviewer's changes were marked in the manuscript in yellow.

Round 2

Reviewer 1 Report (Previous Reviewer 3)

Comments and Suggestions for Authors

Lines 284-289: clarify the meaning of "..had 80.1% IgG antibodies...). Is it referred to protection? 

Comments on the Quality of English Language

Some minor editing of English language is required

Reviewer 2 Report (Previous Reviewer 2)

Comments and Suggestions for Authors

The authors addressed my comments.

This manuscript is a resubmission of an earlier submission. The following is a list of the peer review reports and author responses from that submission.

Round 1

Reviewer 1 Report

Comments and Suggestions for Authors

Major concerns:

The method for detecting neutralizing antibody titer is inaccurate (semi-quantitative) and the result presented is not eligible for comparison. Thus the analysis in Table 2 and 3 are not based on the true data of antibody titer. The major conclusions in this paper are not solid (though many other papers already reached the same conclusions). If you were using this kit (https://www.genscript.com/sars-cov-2-neutralizing-antibody-rapid-test.html), please specify how you convert the test result (colorimetric) to the percentage number shown in Figure 1. The most accepted method for measuring serum nAb titer is with pseudotyped virus, or at least with RBD ELISA, in which serial dilution of serum is required to plot a curve and calculate the NT50 value.

The statistical analysis used in this paper is not accurate either. In Table 5, it is not clearly stated whether you are analyzing patients re-infected with Omicron or any SARS-CoV-2 variants.

Minor points:

Pay attention to significant figures. Should keep 3 digits for all the data presented in the paper. 

Comments on the Quality of English Language

Please revise some wording to make the sentences more concise. (see attachment)

Reviewer 2 Report

Comments and Suggestions for Authors

The manuscript by Dr. Vargas-De León and colleagues is well written and their findings are clearly exposed. However, their study fits into a context now full of similar publications, even more so by bringing data collected after just two vaccination doses. A novel element that may attract readers is the inclusion of risk factors in the study. Still, this aspect should be made more prominent by adding a multivariate analysis and its graph. The points should be addressed before publication:

1. Explain better the neutralization test performed: is it able to detect antibodies directed against the S or N protein, or both? Is it able to detect antibodies against the VOCs circulating during the time of the study and its follow-up? How was this test validated? Please add proper references, as the serological tests developed in the first phases of the pandemic were not accurate (e.g.  DOI: 10.1002/jmv.26605).

2. Add statistics to Figure 1. Are there reported total IgG anti-SARS-CoV-2 or only the neutralizing fraction? Please specify this aspect in both the figure caption and the y-axis of the graph. If the total IgG is reported, how was identified? Please add this in the materials and methods section.

3. Add a multivariate analysis, for example, a correlation matrix in R with the consequent matrix heatmap plot.

Minor:

Line 42: change the font of reference 2, it is different from the other references.
Line 48: update epidemiological data to 2023

Reviewer 3 Report

Comments and Suggestions for Authors

In this work Cruz Vargas – De Leòn et al focused on the neutralizing antibodies specific for SARS-CoV-2 elicited in healthy people vaccinated with mRNA vaccine Pfizer/BioNTech in a Mexican hospital, dissecting the influence of comorbidities on the immune response generated in subject previous infected or not with the virus. Authors demonstrate the effect of previous infection on the generation of persistent neutralizing antibodies, and the negative effect of obesity, alcoholism, and diabetes.

The work refers to immune response after the second vaccine dose, not updated to the SARS-CoV-2 booster immunization. For this reason, I suggest empathizing the point of critical factors that can affect (in general) the immune response, as a tool to monitor health care workers also against other pathogen infections.

-Some points need to be clarified and revised for a better understanding of the statements:

-       - Revise sentence at line 23-25

-        -Define the isotype of antibody in abstract and in the text.

-       - I suppose that ELISA used to test neutralizing response is a surrogate neutralizing test. Please, clarify this concept in materials and methods section.

-        -Ref 2 at line 46 needs to be revised in terms of formatting.

-        -Include new update references on covid vaccines at line 50.

-        -Specify the interval between doses and the time points of analysis in materials and methods.

-        -Detail the collection period of samples, and detail the sample treatment of sera/plasma??

-        -Include inclusion and exclusion criteria.

-        -Detail the name of the protocol for clinical study approved by the ethical committee.

-       - Indicate the meaning of * in Tab 1

-       - Include a line in Tab 1 with the total population of characteristics reported (independently from the viral infection)

-        -Define the unit used to quantify IgG antibodies and include in the text and in y of fig. 1.

-        -Define and clarify in the whole manuscript the meaning of “percentage of antibodies” and clarify the reference.

-        -Include statistic in Fig. 1 and detail in fig. legend.

-        -Avoid specifying Pfizer BioNTech every time in figure legend, since it is the only vaccines included in the work.

-        -Specify what the mean reported is Table 2 is referred.

-        -At line 167, correct COVID-19 with SARS-CoV-2, since it refers to infection.

-        -Define saturated and minimal model in Tab 3

-        -Revise Tab 4 since it is difficult the association of different variables reported. A table using + / - maybe can help the reading.

-        -Revise lines 190-193.

-       - In lines 202-203, indicate the total number of population (ie. 18 and 8 re-infection on the total of…).

-        -In Tab 5, clarify better values reported in brackets.

-        -Clarify line 213-215, and info regards statistical differences.

-        -Define abbreviation in Tab 6.

-        -Specify in conclusion how comorbidities as obesity and diabetes can alter immune response (specify the reduction or the negative effect), and the effect of previous infection in the cohort analyzed.

Comments on the Quality of English Language

Moderate editing of English language required.